# Muscle Glycogen Assessment and Relationship with Body Hydration Status: A Narrative Review

**DOI:** 10.3390/nu15010155

**Published:** 2022-12-29

**Authors:** Keisuke Shiose, Hideyuki Takahashi, Yosuke Yamada

**Affiliations:** 1Faculty of Education, University of Miyazaki, Miyazaki 889-2192, Japan; 2Faculty of Health and Sport Sciences, Advanced Research Initiative for Human High Performance, University of Tsukuba, Tsukuba 305-8577, Japan; 3Department of Physical Activity, National Institutes of Biomedical Innovation, Health and Nutrition, Tokyo 162-8636, Japan

**Keywords:** glycogen, magnetic resonance spectroscopy, body composition, performance

## Abstract

Muscle glycogen is a crucial energy source for exercise, and assessment of muscle glycogen storage contributes to the adequate manipulation of muscle glycogen levels in athletes before and after training and competition. Muscle biopsy is the traditional and gold standard method for measuring muscle glycogen; alternatively, ^13^C magnetic resonance spectroscopy (MRS) has been developed as a reliable and non-invasive method. Furthermore, outcomes of ultrasound and bioimpedance methods have been reported to change in association with muscle glycogen conditions. The physiological mechanisms underlying this activity are assumed to involve a change in water content bound to glycogen; however, the relationship between body water and stored muscle glycogen is inconclusive. In this review, we discuss currently available muscle glycogen assessment methods, focusing on ^13^C MRS. In addition, we consider the involvement of muscle glycogen in changes in body water content and discuss the feasibility of ultrasound and bioimpedance outcomes as indicators of muscle glycogen levels. In relation to changes in body water content associated with muscle glycogen, this review broadens the discussion on changes in body weight and body components other than body water, including fat, during carbohydrate loading. From these discussions, we highlight practical issues regarding muscle glycogen assessment and manipulation in the sports field.

## 1. Background

Muscle glycogen is synthesized from carbohydrates and is stored in the muscles, liver, and brain. Since muscle glycogen is used as the main fuel source of exercise, a significant amount of glycogen is consumed during prolonged or high-intensity exercise. Decreased glycogen levels result in fatigue by impairing sarcoplasmic reticulum Ca^2+^ release [1,2]. Therefore, maintaining muscle glycogen levels through the adequate intake of carbohydrates is important for athletes preparing for hard training or competition. The muscle glycogen levels are supercompensated for approximately 2.0-fold of the preloaded value after a high-carbohydrate diet with exercise tapering for approximately ˃3 days [3]. This approach is widely known as “carbohydrate loading.” It is well known that increasing muscle glycogen level with carbohydrate loading improves endurance performance. Moreover, a new glycogen strategy known as “train low” has been developed in recent years [4,5]. The typical procedure of train low involves the depletion of 40–50% of glycogen levels by prior training, and subsequent training is conducted at a low glycogen state [6]. Training at a low glycogen state enhances molecular metabolic adaptation of muscle cells [7]. In this context, to improve exercise performance, athletes should understand their glycogen level and control it adequately; thus, they must understand how to increase glycogen levels and decrease and maintain low glycogen levels in specific instances, before training.

A muscle biopsy procedure developed by Bergstrom [8] has become the traditional and “gold standard” method to assess muscle glycogen levels. In this procedure, a dedicated needle is inserted mainly in the vastus lateralis, and approximately 20–100 mg of muscle sample is obtained. The muscle glycogen level in the sample is determined via biochemical analysis. However, due to its invasiveness, the biopsy technique is difficult to apply routinely for athletes. Currently, magnetic resonance spectroscopy (MRS) has been developed as a non-invasive and reliable glycogen assessment method [9]. By using an MRS system with a high-strength magnetic field (>3 tesla), muscle glycogen can be briefly measured from the spectra of a naturally occurring stable carbohydrate isotope (^13^C).

It is well known that muscle glycogen accumulation is associated with increased body water content [10,11]. Because body water accumulation may be reflected as a change in pixel intensity of the tissue image using ultrasound technique or electrical resistance using the bioimpedance (BIA) technique, these parameters can potentially be used as indirect parameters for assessing muscle glycogen levels. However, because the relationship between muscle glycogen and body water change remains undetermined, we should carefully consider whether these parameter changes directly reflect glycogen levels or not.

In this mini-review, we present a recently developed method for muscle glycogen assessment focused on ^13^C MRS. In addition, we revisit the relationship between muscle glycogen and body water content and discuss the feasibility of ultrasound and bioimpedance outcomes as indicators of muscle glycogen levels. In this context, we review the changes in body weight and composition during carbohydrate loading and discuss the effectiveness of carbohydrate loading in various types of sports.

## 2. Muscle Glycogen Measurement by ^13^C MRS

^13^C MRS is an attractive method for repeated measurement of glycogen in skeletal muscle, liver, and brain in a wide range of individuals, including women, children, and athletes, because of its non-invasiveness (Figure 1) [11,12,13,14,15]. However, the main disadvantage of ^13^C MRS is the low natural abundance of ^13^C (1.1%) and the resulting relatively low sensitivity. Furthermore, ^13^C MRS requires dedicated hardware and software, making it difficult to perform on a general clinical MR system.

Using an MRS system with higher static magnetic field strengths is crucial to obtaining reliable ^13^C MRS measurements in the shortest possible time. In previous studies using 1.5 T MR systems, a data collection time of 15–30 min with a repetition time (TR) of 180–700 ms and acquisitions of 2500–5000 were required, whereas with a 3 T or 4.7 T MR, acquisitions required 5–15 min with a TR of 75–230 ms and acquisitions of 2700–7800. To obtain more reliable and analyzable spectra with a high signal-to-noise ratio, proton decoupling and nuclear Overhauser enhancement techniques are essential. Previously, a second radiofrequency (RF) channel system, which is not generally installed, was needed to apply these techniques; however, recently, dual-nuclear acquisition and decoupling have become possible with clinical MR systems by switching channels at high speed [16].

Muscle glycogen concentration is determined by comparison with an external standard solution in a cylindrical phantom that replaces the subject’s region (such as the thigh or calf) in an identical position relative to the RF coil. In general, since the localization method is not used, the size of the phantom must be identical to that of the muscle for calculating the absolute concentration. MRI images of the subject’s region are acquired to determine the phantom size, and the subject’s muscle size is measured before ^13^C MRS. The RF coil parameters (tuning, linewidth, and noise) do not change when switching from the subject’s region to the phantom, and this is considered a verification that coil loading remains constant [17,18]. To calibrate the ^13^C pulse, a known concentration of ^13^C solution, such as sodium acetate, acetone, and formic acid, is used, which is positioned in the center of the surface coil. Moreover, when comparing the glycogen peak areas between the muscle and the phantom, it is necessary to consider the effects of temperature. Because the peak area of the standard solution increases proportionally with temperature increase (unpublished data), a correction equation is applied to calibrate the area.

The absolute glycogen concentration obtained using ^13^C MRS overlaps with the values obtained through the biochemical analysis of muscle samples [17]. The coefficient of variation in repeated measurements of muscle glycogen using ^13^C MRS with repositioning is 3.5%–10% [19]. Although the ^13^C MRS is a reliable method for measuring muscle glycogen, the results should be interpreted considering this range of variability.

## 3. Feasibility of Ultrasound and Bioimpedance Outcomes as Indicators of Muscle Glycogen Levels

Since athletes participate in training and competitions at various locations, developing a portable and easy method to assess muscle glycogen content is desirable. In this regard, the ultrasound method is potentially a readily available glycogen assessment method in the field. This method was clinically developed in the mid-20th century and was applied to the skeletal muscles to diagnose muscle-related diseases and measure muscle cross-sectional area or thickness. Regarding its association with glycogen, Tuthill et al. initially showed that the ultrasound attenuation coefficients change with the liver glycogen level in ex vivo and in vivo experiments [20]. Furthermore, an ultrasound image analysis system was recently developed to assess muscle glycogen content (Musclesound^®^ system, MuscleSound Inc., Denver, CO, USA) [21]. This system assesses muscle glycogen levels as the “glycogen score,” which is likely to be calculated based on the pixel intensity of the ultrasound image [21,22]. A study using this system in competitive cyclists suggested that the glycogen score is correlated with muscle glycogen levels determined using the biopsy technique [21,22]. Furthermore, a decrease in the glycogen score after a soccer match has been reported; the authors argued that the decrease in the glycogen score reflects the decrease in muscle glycogen content [23]. Changes in ultrasound outcomes were believed to be mainly due to body fluid shifts by the increase or decrease in glycogen stores [20,21,22]. However, a study by another group indicated that muscle water content might not have been matched to muscle glycogen status after exercise and recovery periods [24]. In addition, it was demonstrated that no relationship was observed between the glycogen score and the muscle glycogen content using the MuscleSound^®^ system and biopsy technique, respectively [24,25]. They claimed that the ultrasound technique, which is based on pixel intensity measurement, is not valid for muscle glycogen assessment [24,25,26].

Electrical resistance, measured using the bioimpedance analysis (BIA) method, may be associated with altered muscle glycogen levels. In the BIA method, an imperceptible current is applied to the whole body or a specific body segment, and electrical resistance components are measured easily and non-invasively. Although current flow in the body is complicated, it is considered—in a simplified equivalent circuit model—that low-frequency current flows through the extracellular space of tissue, whereas high-frequency current flows through both the intracellular and extracellular spaces since the cell membrane acts as a capacitor. Therefore, BIA using multifrequency currents can separately detect the electrical resistance of intracellular and extracellular components (Figure 2) [27,28]. Body electrical resistance is mainly determined through body hydration status, and a change in BIA is basically a refractive change in body water content.

Our recent study using BIA with multifrequency current indicated that the electrical resistance of the intracellular component decreases when muscle glycogen is supercompensated in a well-controlled laboratory setting [11]. Another study revealed that carbohydrate loading induces a possible clear change in intracellular water and a likely clear change in extracellular water using BIA [29]. This means that electrical resistance is altered with carbohydrate loading. In glycogen depletion conditions, electrical resistance was reported to either increase [29] or remain constant [30]. These results indicate the possibility that body electrical resistance could be an indicator of muscle glycogen accumulation. It is currently established that changes in muscle glycogen levels and electrical resistance occur simultaneously; however, a direct relationship between these changes remains unknown. Moreover, the electrical resistance of a body has also been suggested to be altered by factors that are relatively difficult to control in the field: for example, changes in metabolite levels other than glycogen [31], ambient air temperature [32], and measuring time [33]. Therefore, further studies are required to clarify the feasibility of the BIA method as an indicator of glycogen state in the field.

Both ultrasound and BIA methods are suitable for field application since they are easy to perform, have lower costs, and are portable. However, as mentioned above, applying these methods for muscle glycogen assessment currently appears difficult. Therefore, more suitable techniques should be developed to establish assessment techniques for muscle glycogen in the field.

## 4. Muscle Glycogen and Body Hydration Status

An increase in stored glycogen is believed to be accompanied by an increase in body water content since glycogen is highly hydrophilic. Regarding the water bound to glycogen, Zants et al. initially estimated that 1 g of liver glycogen binds 3 g of water using the data provided by Pavy [34,35]. Subsequent studies showing the association between glycogen and body water are summarized in Table 1. Numerous animal studies assessing liver glycogen and water content indicated that 1 g of liver glycogen is bound to 1.6–3.8 g of water [36,37,38,39].

Concerning muscle glycogen in humans, Olsson and Saltin initially suggested that 1 g of muscle glycogen is bound to 3–4 g of water, based on the results of the tritium dilution method for detecting body water content and sampled using muscle biopsy for muscle glycogen content under glycogen-depleted and loaded conditions [10]. Furthermore, a study using magnetic resonance imaging presumed that intracellular water binding increases in glycogen-loaded muscles [47]. Our previous study using the deuterium dilution method for measuring total body water content also showed that body water content increased in glycogen-loaded conditions; the segmental BIA method indicated that the main component of increased body water was intracellular water (Figure 3) [11].

However, the relationship between body water and stored glycogen is inconclusive. As mentioned above, human studies have shown an increase in body water in supercompensated glycogen conditions; however, results obtained from animal studies are confusing. First, some studies have confirmed a significant relationship with liver glycogen [36,37,38,39,41,42,44], whereas other studies did not confirm this relationship with liver glycogen [40,43] and muscle glycogen [41,45,46]. These controversial results are possibly due to a change in the size and number of glycogen molecules. Glycogen is formed by chains of glucose residues, and both the size and amount of glycogen may change with diet and exercise [49]. Although the respective effect of glycogen size and amount on bound water content was not revealed, a complicated change in glycogen structure may make it difficult to predict the relationship between muscle glycogen and bound water. Notably, a recent study indicated that glycogen supercompensation in humans increases glycogen number rather than size [50]. Second, body water flow, which was not associated with glycogen, should be considered. Fernández Elas et al. investigated the muscle glycogen resynthesis and muscle water content in humans after exercise using the biopsy technique. After exercise, a 400 mL or ~3170 mL drink, which included equivalent amounts of carbohydrate, was ingested. The ratio of stored glycogen and water in muscle changed from 1:3 to 1:17 in response to the amount of fluid intake. The authors assumed from this result that 1 g of muscle glycogen is stored with 3 g of water; however, water, which is not associated with glycogen, may also be stored depending on the amount of fluid provided [48].

Consequently, although inconsistent results were shown in animals, it can be assumed that body water increases approximately 3–4-fold when muscle glycogen is stored in humans. However, it should be noted that, since glycogen-binding water was not directly measured, the increase in body water accompanied by muscle glycogen storage cannot be concluded to be actually due to glycogen-binding water. Water not associated with muscle glycogen is potentially stored in muscle in response to fluid intake and may alter the stored body water and muscle glycogen ratio.

## 5. Change in Body Composition during Carbohydrate Loading

After a few days of a high-carbohydrate diet (carbohydrate 8–12 g/kg/day), an increase in stored muscle glycogen of approximately 1.5–2.0-fold the normal level is observed. In the classical carbohydrate loading procedure, muscle glycogen is initially depleted by exercise and a high-carbohydrate diet is consumed for 3 days following 2–3 days of a low-carbohydrate diet [51]. Interestingly, improved procedures have been described in subsequent studies [52,53]. A review article by Burke et al. provides good details [54].

During carbohydrate loading, it is generally understood that body weight increases by approximately 1.0–1.5 kg, mainly due to an increase in body water content, as mentioned in the previous section. However, changes in other body components are uncertain, although drastic changes in energy intake and macronutrient balance occur. Michalczyk et al. reported that body weight and lean body mass increased by approximately 1.5 kg, but fat mass was unchanged after a 7-day carbohydrate loading period with an isocaloric high carbohydrate diet in a basketball player using the BIA method [55]. Furthermore, Bone et al. reported that total body and segmental lean mass increased by approximately 1.5 kg, whereas fat mass and bone mineral density remained unchanged during the 5-day carbohydrate loading period with the excess energy diet using the dual X-ray absorptive method [29]. Our study measuring total body water content and body density using the 3-compartment body composition model suggested that carbohydrate loading with excess energy intake (carbohydrate12g/kg/day; 1597 kcal [6682 kJ]/day higher than the daily diet) increases fat-free dry solid (e.g., glycogen and bone) and total body water content, and slightly but significantly decreases fat mass by −0.5 kg (Figure 4) [11].

Simultaneously, we observed a segment-specific increase in intracellular water content that occurred after carbohydrate loading [11]. Therefore, changes in body composition during carbohydrate loading may have the following features: (1) whole body or segment-specific increase in fat-free mass, which is mainly due to an increase in body water, and (2) no increase in fat mass even though the amount of energy intake is temporarily more than the daily intake. Note, we should understand the possibility that change in water distribution after carbohydrate loading induces an estimation error for body composition assessment because the method assumes a constant tissue hydration state for calculating body composition [11,29,30].

However, because only a few studies have assessed changes in body weight and composition during carbohydrate loading, the effects of these changes on performance are unclear. Interestingly, some bodybuilders likely engage in carbohydrate loading before competitions to increase their muscle volume and physical appearance because they believe that the body water is stored in the muscles [56,57,58]. In an experimental study, Moraes et al. suggested that carbohydrate loading may have contributed to an acute increase in muscle volume and physical appearance in male bodybuilders based on changes in body weight and muscle thickness and circumferences [59]. On the other hand, it is suspected that weight gain after carbohydrate loading can increase physical load during exercise [60,61]. Madsen et al. reported that the running economy at 75%–80% VO_2_max intensity in elite endurance runners remained unchanged after 3 days of a high carbohydrate diet (carbohydrates accounted for 70% of energy intake); however, change in body weight and composition were not measured [62]. Accordingly, additional studies on the effects of body weight and composition changes should be conducted to provide insight into the effectiveness of carbohydrate loading in athletes.

## 6. Conclusions

To assess muscle glycogen storage, alternative non-invasive methods to muscle biopsy have been developed. ^13^C MRS is the most reliable alternative method to measure muscle glycogen levels. In the field, ultrasound and BIA outcomes, which reflect body hydration status, have the potential to be used as indicators of muscle glycogen levels. However, currently, reliable conclusions on the use of these methods for muscle glycogen assessment cannot be made due to the uncertain relationship between muscle glycogen level and body hydration status. Regarding the change in body composition during carbohydrate loading, although fat mass does not increase, weight increases, which is mainly due to increased body water content. Despite these findings, studies on body weight and composition changes during carbohydrate loading and their effects on performance are still insufficient. The positive and negative aspects of carbohydrate loading should be further studied to expand its applicability.

## Figures and Tables

**Figure 1 nutrients-15-00155-f001:**
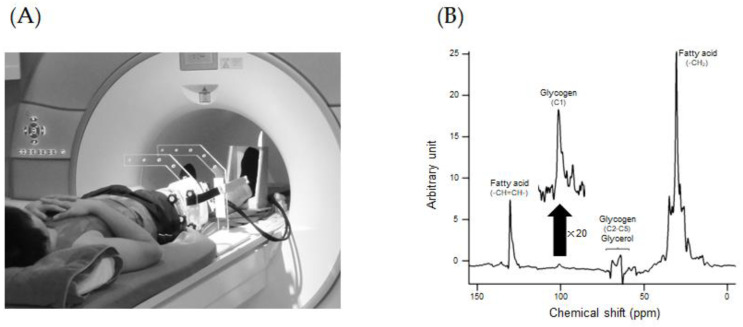
View of ^13^C MRS measurement of the thigh muscle (**A**) and representative ^13^C MRS spectrum of the muscle (**B**).

**Figure 2 nutrients-15-00155-f002:**
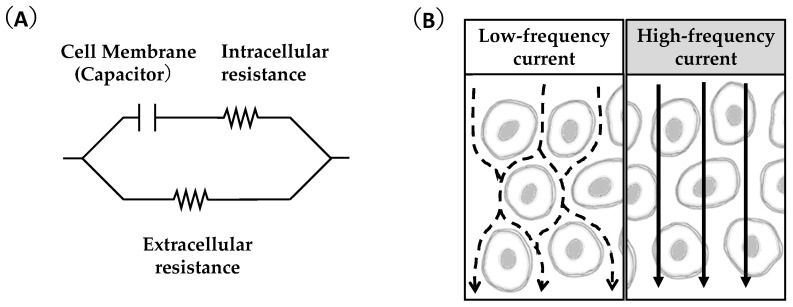
Equivalent circuit model of multi-frequency bioimpedance analysis (**A**) and the flows of low- and high-frequency currents (**B**) for a tissue.

**Figure 3 nutrients-15-00155-f003:**
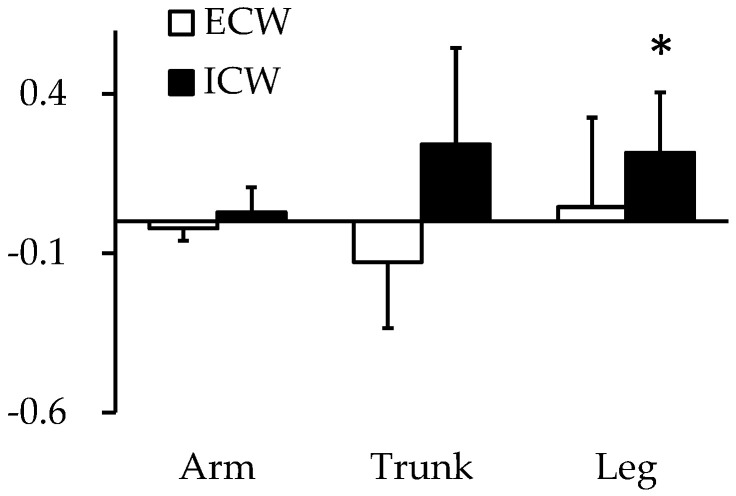
Change in water distribution during carbohydrate loading assessed using segmental BIA. Values are expressed as mean ± standard deviation of delta value (post-loaded value − pre-loaded value). ECW, extracellular water; ICW, intracellular water. * Significant (*p* < 0.05) difference between pre-loaded value and post-loaded value.

**Figure 4 nutrients-15-00155-f004:**
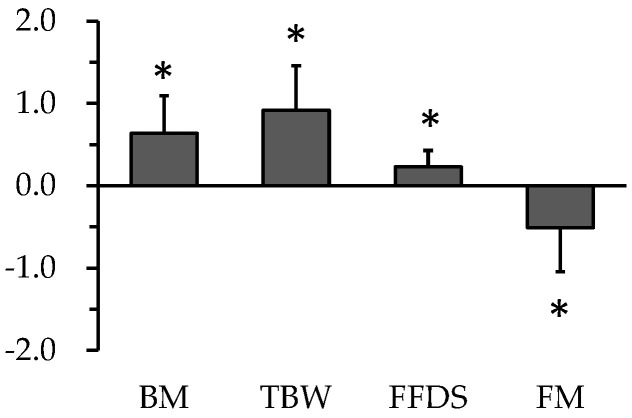
Change in body composition during carbohydrate loading. Values are expressed as mean ± SD of delta value (post-loaded value − pre-loaded value). BM, body mass; TBW, total body water; FFDS, fat-free dry solid; FM, fat-mass. * Significant (*p* < 0.05) difference between pre-loaded value and post-loaded value.

**Table 1 nutrients-15-00155-t001:** Studies involving muscle or liver glycogen and body water content.

**Study**	Species	Glycogen Level	Glycogen Assessment		Body Water Assessment		Assessment Timing	Positive Relationship between Glycogen and Water	Estimated Glycogen: Water Ratio
			**Organ**	**Method**	**Organ**	**Method**			
Bridge and Bridges [40]	Rabbits	Low to high	Liver	Biochemical analysis	Liver	Dried and weighed	Post dietary manipulation	No	
MacKay and Bergman [41]	Rabbits	Low to high	(1) Liver (2) Muscle	Biochemical analysis	(1) Liver (2) Muscle	Dried and weighed	Post dietary manipulation	Liver; Yes Muscle; No	
Puckett and Wiley [36]	Rats	Low to high	Liver	Biochemical analysis	Liver	Dried and weighed	Post dietary manipulation	Yes	1:2.4
MacKay and Bergman [37]	Rats	Low to high	Liver	Biochemical analysis	Liver	Dried and weighed	Post dietary manipulation	Yes	1:3.8
Greisheimer and Goldsworthy [42]	Rats	Low to high	Liver	Biochemical analysis	Liver	Dried and weighed	Post dietary manipulation	Yes (Only in a condition where the rats were given a diet without priori fasting,)	
Kaplan and Chaikoff [43]	Dogs	Low to high	Liver	Biochemical analysis	Liver	Weighted total lipid and defatted dried tissues.	Post dietary manipulation	No	
Fenn [44]	Rats	Low, normal, high	Liver	Biochemical analysis	Liver	Dried and weighed	Post dietary manipulation	Yes	
Fenn and Haege [38]	Cats	Low to high	Liver	Biochemical analysis	Liver	Dried and weighed	Post dietary manipulation	Yes	1: 1.63
McBride et al. [39]	Rats	Low to high	Liver	Biochemical analysis	Liver	Dried and weighed	Post dietary manipulation	Yes	1:2.7
Olsson and Saltin [10]	19 young healthy males	Low to high	Muscle (biopsy sampling)	Biochemical analysis	Whole-body	IDM	(1) Post-protein and fat diet (2) Post-carbohydrate and protein diet	Yes	1:3–1:4
Sherman et al. [45]	Rats	Low to high	Muscle	Biochemical analysis	Muscle	Dried and weighed	Post dietary manipulation	No	
Richter et al. [46]	Rats	Normal to high	Muscle	Biochemical analysis	Muscle	Dried and weighed	Post perfusion	No	
Nygren et al. [47]	5 healthy males	Low to high	Muscle (biopsy sampling)	Biochemical analysis	Muscle	Magnetic resonance imaging	(1) Post carbohydrate-restricted diet (2) Post high-carbohydrate diet	Yes	
Fernández-Elías et al. [48]	9 endurance-trained male cyclists	Low to high	Muscle	Biochemical analysis	Muscle	Dried and weighed	(1) Pre-exercise (2) Post-exercise (3) Post recovery	Yes	1:3–1:17
Shiose et al. [11]	8 healthy males	Normal to high	Muscle	^13^C-MRS	(1) Whole-body (2) Each body segment	(1) IDM (2) BIA	(1) Pre-exercise (2) Post high-carbohydrate diet	Yes	≤1:4
Bone et al. [29]	18 well-trained male cyclists	Low to high	Muscle (biopsy sampling)	Biochemical analysis	Whole-body	BIA	(1) Pre exercise (2) Post exercise (3) Creatin loaded (4) Glycogen loaded (5) Creatin-glycogen loaded	Yes	
Shiose et al. [30]	12 healthy males	Low to normal	Muscle	^13^C-MRS	(1) Whole-body (2) Each body segment	(1) IDM (2) BIA	(1) Pre exercise (2) Post recovery	No	

IDM, isotope dilution method, BIA, bioimpedance analysis, ^13^C-MRS; carbon-13 nuclear magnetic resonance spectroscopy.

## Data Availability

Not applicable.

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
