# Peer review of "Muscle Glycogen Assessment and Relationship with Body Hydration Status: A Narrative Review"

_nutrients, 2022, doi:10.3390/nu15010155_

Round 1

Reviewer 1 Report

This review provided comprehensive details about the methodologies that are being used in measuring the glycogen content in muscle especially focusing on 13C MRS. Further, they discussed the correlation between altered glycogen metabolism and changes in body water.

Author Response

Response to Reviewer 1.

Thank you for your encouraging comments. We appreciate your having carefully read our manuscript.

Reviewer 2 Report

I do not think the present review is within the scope of the journal. The scope of the journal states:

  Nutrients will consider manuscripts for publication that provide novel insights into the impacts of nutrition on human health or novel methods for assessing nutritional status. This includes manuscripts describing the outcomes of animal studies that have relevance to human health.  

I do not think the paper fits the scope because based on a technique.

Author Response

Responses to the comments of Reviewer 2

Thank you for your comments and for reading our manuscript. We agree that this study mentions about the glycogen assessment techniques; however, it also describes the changes in body composition during carbohydrate loading and the resulting effects on exercise performance. Therefore, we believe that this paper will be of interest to athletes and sports nutritionist. 

Reviewer 3 Report

Nutrients-2031574

Title: Muscle glycogen assessment and relationship with body hydration status

General Comments:

This review article discusses currently available muscle glycogen assessment methods focusing on 13C MRS. The review also considers the involvement of muscle glycogen in changes in body water content and discusses the feasibility of ultrasound and bioimpedance outcomes as indicators of muscle glycogen levels. Review of the literature into these topics is important due to the relative ease and non-invasive nature of these assessments techniques as compared to muscle biopsies. Lastly, the review discusses changes in body weight and body components other than body water, including fat, during carbohydrate loading and discusses the effectiveness of carbohydrate loading in various sports. This area is of interest to athletes ranging from endurance sports to physique sports.

The authors did a fairly good job in covering some of the areas if this nuances topic, but I feel they need to dive deeper into the literature and provide more information than is currently provided. Also, I recommend they do a better job tying the pieces together. Also, please address my specific comments below.

Specific Comments:

1.       Title should state “a narrative review”

2.       In background, you introduce 2 ways to carb loading. However, there are others. Can you describe at least one more method in brief.

3.       Remove double space between “low” and “levels” page 2 ln 47.

4.       Page 2 ln 52. Add the word “the” after the “However,”

5.       I am still in the background and I have found a couple of missed words such as “the” or “a” before certain words. I also saw a spacing error. I do not want to correct each of these oversights, but please review the manuscript and fix throughout. I will not point these out any further unless I see major flaws in grammar, spelling, or sentence structure.

6.       Page 2 line 70 “Because 13C MRS is non-invasive, it is an attractive method for repeated measurement of glycogen in skeletal muscle, liver, and brain in a wide range of subjects, including women, children, and athletes” please provide reference.

7.       Page 3 ln 87 “In the past, a second radiofrequency (RF) channel system was needed to apply these techniques, but recently, they have become possible with a broad-band single-RF channel system by switching channels at high speed.” Please provide reference.

8.       Pg. 3 ln 100- “Moreover, when comparing the glycogen peak areas between the muscle and the phantom, it is necessary to consider the effects of temperature. Because the peak area of the standard solution increases in proportion to the increase in temperature, a correction equation is applied to calibrate the area.” Please provide reference.

9.       Page 9 ln 109-115. This paragraph is fine, but what does the reader take away from this? Is one better than the other? Can we compare the results of one methodology to those of another? Do we have larger errors in one vs another? You also need to add another paragraph in this section with a “take home” message for the reader based on the evidence.

1  Pg 3 ln 128-132 “However, study conducted by other groups indicated that muscle water content may have not been matched to muscle glycogen status after exercise and recovery periods and that there was no relationship between the ultrasound outcomes using the MuscleSound® system and the muscle glycogen content using the biopsy technique” Please break this into 2 sentences to improved flow.

1   Page 3 ln 138 “injected” is not the correct word here. We do not inject current. Please modify.

1   The sections on measuring glycogen content with US and BIA seem very vague. Please provide insight from a few more studies to discuss this further with US and BIA.

1   Page 4 ln 154-155. Who observed it? Zants et al or Pavy et al? Please  clarify.

1   Pg 5 ln 184 please rephrase “Fernández Elas et al. investigated muscle glycogen resynthesis after exercise and muscle water content in humans using the biopsy technique.” I think what you mean to say is “Fernández Elas et al. investigated muscle glycogen resynthesis and muscle water content in humans after exercise using the biopsy technique”

1   Page 5 ln 218 “(carbohydrate 12g/lg/day)” I think you mean “(carbohydrate 12g/kg/day)”

1  Page 6 ln 231 “Note, we should understand the possibility that change in water distribution after carbohydrate loading induces an estimation error for body composition assessment in that the method assumes the constant tissue hydration state for the calculation of body composition.” Please provide reference.

1   Pg 6 ln 235. “It is widely accepted that carbohydrate loading improves endurance performance. In a previous review, both the time to exhaustion and the performance of the time trial were shown to improve, especially in cases where the exercise test lasted more than 90 minutes and was improved by carbohydrate loading”. Please break this into 2 sentences for better flow.

1  Pg. 6 ln 248 “Furthermore, Beis et al. showed that in well trained endurance runners a 0.9 kg increase in body weight mainly consisted of body water caused by ingestion of creatinine and glycerol and does not change the running economy when running at 60% VO2max”. You mean creatine, not creatine. Also, what type of creatine Was consumed? Monohydrate? HCL? Ethyl ester or ???

1  You may want to explore the literature on the  effects of CHO loading in other sports such as physique/bodybuilding competitions and integrate that into this section.

Author Response

Responses to the comments of Reviewer 3

We wish to express our appreciation for your insightful comments on our paper. The comments have helped us significantly improve the paper.

Comment 1.

Title should state “a narrative review”

Response 1.

Thank you for this valuable suggestion. Accordingly, we revised the title as “Muscle glycogen assessment and relationship with body hydration status: a narrative review”

Comment 2.

In background, you introduce 2 ways to carb loading. However, there are others. Can you describe at least one more method in brief.

Response 2.

Thank you for this suggestion. In the Background section, we introduce one traditional carb loading method proposed by Bergstrom [3] (redlined manuscript; p.1, lines 36–38). If necessary, we can add other carb loading methods; however, a brief description of carb loading methods was also mentioned at the beginning of section 5, “Change in body composition and performance during carbohydrate loading”

Comment 3.

Remove double space between “low” and “levels” page 2 ln 47.

Response 3.

Thank you for pointing this out. This error has been corrected in accordance with your comment (redlined manuscript; p. 2, line 47).

Comment 4.

Page 2 ln 52. Add the word “the” after the “However,”

Response 4.

Thank you for pointing this out. Accordingly, we corrected this inadvertent error (redlined manuscript; p. 2, line 52).

Comment 5.

I am still in the background and I have found a couple of missed words such as “the” or “a” before certain words. I also saw a spacing error. I do not want to correct each of these oversights, but please review the manuscript and fix throughout. I will not point these out any further unless I see major flaws in grammar, spelling, or sentence structure.

Response 5.

Apologies for such errors. Accordingly, the manuscript has been carefully rechecked, and an English Language proofreading service has reviewed it.

Comment 6.

Page 2 line 70 “Because 13C MRS is non-invasive, it is an attractive method for repeated measurement of glycogen in skeletal muscle, liver, and brain in a wide range of subjects, including women, children, and athletes” please provide reference.

Response 6.

Thank you for pointing this out. Accordingly, we have added references to the section referred to (redlined manuscript; p. 2, lines 76).

Comment 7.

Page 3 ln 87 “In the past, a second radiofrequency (RF) channel system was needed to apply these techniques, but recently, they have become possible with a broad-band single-RF channel system by switching channels at high speed.” Please provide reference.

Response 7.

Thank you for pointing this out. We have revised this statement and added references as follows.

Previously, a second radiofrequency (RF) channel system, which is not generally installed, was needed to apply these techniques; however, recently, deal-nuclear acquisition and decoupling have become possible with a clinical MR system by switching channels at high speed [16] (redlined manuscript; p. 3, lines 90–93).

Comment 8.

Pg. 3 ln 100- “Moreover, when comparing the glycogen peak areas between the muscle and the phantom, it is necessary to consider the effects of temperature. Because the peak area of the standard solution increases in proportion to the increase in temperature, a correction equation is applied to calibrate the area.” Please provide reference.

Response 8.

Thank you for your suggestion. We do have robust evidence about glycogen peak area and temperature of the glycogen solution; however, this has not been published yet. Therefore, we added “unpublish data” to the corresponding section (redlined manuscript; p. 3, line 109).

Comment 9.

Is one better than the other? Can we compare the results of one methodology to those of another? Do we have larger errors in one vs another? You also need to add another paragraph in this section with a “take home” message for the reader based on the evidence.

Response 9.

Thank you for your comment. In light of the overall structure and content of the paper, this paragraph is considered unnecessary and has been deleted (redlined manuscript; p.3, lines 117–123).

Comment 10.

Pg 3 ln 128-132 “However, study conducted by other groups indicated that muscle water content may have not been matched to muscle glycogen status after exercise and recovery periods and that there was no relationship between the ultrasound outcomes using the MuscleSound® system and the muscle glycogen content using the biopsy technique” Please break this into 2 sentences to improved flow.

Response 10.

Thank you for this suggestion. Accordingly, we have revised this sentence (redlined manuscript; p. 4, lines 150–155).

Comment 11.

Page 3 ln 138 “injected” is not the correct word here. We do not inject current. Please modify.

Response 11.

Thank you for pointing this out. Accordingly, we corrected this sentence (redlined manuscript; p. 4, line 164).

Comment 12.

The sections on measuring glycogen content with US and BIA seem very vague. Please provide insight from a few more studies to discuss this further with US and BIA.

Response 12.

We appreciate your comment on this point. Accordingly, we revised this section and tried to add some new insight to this section (redlined manuscript; p. 4, line 124–p.5, line 207).

Comment 13.

Page 4 ln 154-155. Who observed it? Zants et al or Pavy et al? Please clarify.

Response 13.

Thank you for your question. Accordingly, we have made the following revision to this text:

“Regarding the water bound to glycogen, Zants et al. initially estimated that 1 g of liver glycogen binds 3 g of water from the data provided by Pavy [34,35]” (redlined manuscript; p. 5, lines 214–216).

Comment 14.

Pg 5 ln 184 please rephrase “Fernández Elas et al. investigated muscle glycogen resynthesis after exercise and muscle water content in humans using the biopsy technique.” I think what you mean to say is “Fernández Elas et al. investigated muscle glycogen resynthesis and muscle water content in humans after exercise using the biopsy technique”

Response 14.

This sentence has been revised according to your suggestion (redlined manuscript; p. 8, lines 255–258).

Comment 15.

Page 5 ln 218 “(carbohydrate 12g/lg/day)” I think you mean “(carbohydrate 12g/kg/day)”

Response 15.

Thank you for pointing this out. This error has been corrected in accordance with your comment (redlined manuscript; p. 9, line 290).

Comment 16.

Page 6 ln 231 “Note, we should understand the possibility that change in water distribution after carbohydrate loading induces an estimation error for body composition assessment in that the method assumes the constant tissue hydration state for the calculation of body composition.” Please provide reference.

Response 16.

Thank you for this suggestion. Accordingly, we have added references for the text referred to (redlined manuscript; p. 9, line 306).

Comment 17.

Pg 6 ln 235. “It is widely accepted that carbohydrate loading improves endurance performance. In a previous review, both the time to exhaustion and the performance of the time trial were shown to improve, especially in cases where the exercise test lasted more than 90 minutes and was improved by carbohydrate loading”. Please break this into 2 sentences for better flow.

Response 17.

Thank you for this suggestion. We have made appropriate revision based on your suggestion (redlined manuscript; p. 10, lines 321–325).

Comment 18.

Pg. 6 ln 248 “Furthermore, Beis et al. showed that in well trained endurance runners a 0.9 kg increase in body weight mainly consisted of body water caused by ingestion of creatinine and glycerol and does not change the running economy when running at 60% VO2max”. You mean creatine, not creatine. Also, what type of creatine Was consumed? Monohydrate? HCL? Ethyl ester or ???

Response 18

Apologies for this error. You are right. They used monohydrate creatine, not creatinine. This error has been corrected (redlined manuscript; p. 10, lines 334–338).

Comment 19.

You may want to explore the literature on the effects of CHO loading in other sports such as physique/bodybuilding competitions and integrate that into this section.

Response 19.

Thank you for your suggestion. We agree with your suggestion that additional information on the effects of carbohydrate loading in other sports would be valuable. Therefore, we added the effects of carbohydrate loading in bodybuilders in the section 5 titled “Change in body composition and performance during carbohydrate loading” (redlined manuscript; p. 10, lines 307–316).

Round 2

Reviewer 2 Report

The review has improved. However, Figure 2A shows a very limited representation of the circuit which could be elaborated for a tissue, considering the complexity of resistance contributors.

Author Response

Comment of Reviewer 2

The review has improved. However, Figure 2A shows a very limited representation of the circuit which could be elaborated for a tissue, considering the complexity of resistance contributors.

Response to Reviewer 2

Thank you for this suggestion. Accordingly, the sections involved with Figure 2A and the title of Figure 2 have been revised (redlined manuscript; p. 5, line 198–202; Figure 2).

Reviewer 3 Report

Thank you for addressing my comments. Before fully accepting, minor flaws in some sentences still exist. You need to do a complete and thorough grammatical review for these minor issues.

monohydrate creatine should read creatine monohydrate

You made good additions to the paper to address my content concerns

Author Response

Comment of Reviewer 3

Thank you for addressing my comments. Before fully accepting, minor flaws in some sentences still exist. You need to do a complete and thorough grammatical review for these minor issues.

monohydrate creatine should read creatine monohydrate

You made good additions to the paper to address my content concerns

Response to Reviewer 3

Thank you for this suggestion. Accordingly, the suggested point has been revised (redlined manuscript; p. 12, line 388), and the English Language proofreading service has reviewed all parts of the manuscript.